Overexpression of AmCBF1 enhances drought and cold stress tolerance, and improves photosynthesis in transgenic cotton

Lu Guoqing 1 2
Wang Lihua 1
Zhou Lili 1
Su Xiaofeng 1
Guo Huiming guohuiming@caas.cn 1
Cheng Hongmei chenghongmei@caas.cn 1
1 Chinese Academy of Agricultural Sciences, Biotechnology Research Institute , Beijing , China
2 Tianjin Academy of Agricultural Sciences, Institute of Germplasm Resources and Biotechnology , Tianjin , China
Mostofa Mohammad Golam
Electronic publication date: 2022 May 25
Publication date: 2022
Volume: 10
Electronic Location ID: e13422
Received 2021 Nov 30; Accepted 2022 Apr 20
Copyright: ©2022 Lu et al.
Copyright year: 2022
Copyright holder: Lu et al.
License: This is an open access article distributed under the terms of the Creative Commons Attribution License, which permits unrestricted use, distribution, reproduction and adaptation in any medium and for any purpose provided that it is properly attributed. For attribution, the original author(s), title, publication source (PeerJ) and either DOI or URL of the article must be cited.
License URL: https://creativecommons.org/licenses/by/4.0/

Keywords: AmCBF1, Gene expression, Transgenic cotton, Stress tolerance, Photosynthesis

Funding: Project for Breeding Genetic Modified Organisms, China 2020ZX08009-05B Project of the Innovation Team Building in Key Areas of Xinjiang Production and Construction Corps (XPCC) 2019CB008 Central Public-interest Scientific Institution Basal Research Fund Y2022PT22 This work was supported by the Key Project for Breeding Genetic Modified Organisms, China (No. 2020ZX08009-05B), the Project of the Innovation Team Building in Key Areas of Xinjiang Production and Construction Corps (XPCC) (2019CB008), and the Central Public-interest Scientific Institution Basal Research Fund (No. Y2022PT22). The funders had no role in study design, data collection and analysis, decision to publish, or preparation of the manuscript.

==============================
China’s main cotton production area is located in the northwest where abiotic stresses, particularly cold and drought, have serious effects on cotton production. In this study, Ammopiptanthus mongolicus C-repeat-binding factor (AmCBF1) isolated from the shrub Ammopiptanthus mongolicus was inserted into upland cotton (Gossypium hirsutum L.) cultivar R15 to evaluate the potential benefits of this gene. Two transgenic lines were selected, and the transgene insertion site was identified using whole-genome sequencing. The results showed that AmCBF1 was incorporated into the cotton genome as a single copy. Transgenic plants had distinctly higher relative water content (RWC), chlorophyll content, soluble sugar content, and lower ion leakage than R15 after drought and cold stress. Some characteristics, such as the area of lower epidermal cells, stomatal density, and root to shoot ratio, varied significantly between transgenic cotton lines and R15. Although the photosynthetic ability of transgenic plants was inhibited after stress, the net photosynthetic rate, stomatal conductance, and transpiration rate in transgenic plants were significantly higher than in R15. This suggested that an enhanced stress tolerance and photosynthesis of transgenic cotton was achieved by overexpressing AmCBF1. All together, our results demonstrate that the new transgenic cotton germplasm has great application value against abiotic stresses, especially in the northwest inland area of China.

Introduction

The growth and yield of cotton (Gossypium hirsutum L.), an important commercial crop worldwide (Zhu et al., 2017), is severely compromised by drought, low temperature, and other abiotic stresses (Hao et al., 2018). Therefore, improving stress tolerance in cotton is an increasingly critical objective.

In China, the northwest inland area (Xinjiang cotton growing region) has become the largest cotton growth and production area (Feng et al., 2017; Li et al., 2010; Zhang et al., 2016). Cotton production has greatly promoted economic development in the harsh arid and semi-arid region of Xinjiang (Jin & Xu, 2012; Wang et al., 2018), which is characterized by major seasonal water shortages, low temperatures, frost, and frequent natural disasters (Feng et al., 2017; Shen et al., 2013). Such conditions can cause serious molecular, morphological, physiological, and biochemical changes in plants, alter normal root development and damage cells, render seedlings more susceptible to diseases, and ultimately reduce crop yield (Wang, Luttge & Ratajczak, 2001; Dai & Dong, 2014).

The transduction of perceived stress signals in plants activates various physiological and metabolic responses, including the expression of stress-responsive genes (Tran et al., 2004). Drought and low temperatures can reduce photosynthesis and ultimately lead to premature senescence of cotton (Chen & Dong, 2016). Drought and water shortage can reduce leaf area, dry matter accumulation, boll number, and boll weight (Zhang et al., 2016; Gerik et al., 1996). Low temperature and drought during sowing and soil salinity stress usually decrease the rate of seed emergence and increase seedlings’ susceptibility to disease (Dai & Dong, 2014).

A large number of genes have been identified as having the potential to increase drought and cold tolerance through genetic engineering (Yue et al., 2012; Verma et al., 2019; Shi et al., 2020; Su et al., 2020). C-repeat binding transcription factors/dehydration responsive element binding transcription factors (CBF/DREB) play a vital role in plant response and resistance to abiotic stresses including cold, drought, and salt (Gilmour et al., 1998; Haake et al., 2002; Ito et al., 2006; Zhang et al., 2017; Moon et al., 2019; Zhou et al., 2020). Using the yeast one-hybrid screening and homologous cloning method, the homologous gene of CBF has been cloned from maize, barley, and other plants (Wang et al., 2008; Skinner et al., 2005). In cotton, 21 CBF genes (GhCBF1-GhCBF21) were divided into four groups based on phylogenetic analysis and were observed to be involved in the cold stress response (Ma et al., 2014). CBF/DREB proteins contain a conserved AP2/EREBP domain, specifically bind to drought-responsive elements (DRE), regulate the expression of downstream stress-related genes, and improve stress tolerance in transgenic plants. Cold and drought tolerance has been conferred to numerous plants by the ectopic expression of the CBF gene. For example, the expression of CBF1 driven by the CaMV 35S promoter in tomato increased drought resistance and the resistant trait was stably inherited (Hsieh et al., 2002). Overexpression of CBF1 and CBF2 in Arabidopsis thaliana increased tolerance to freezing and water stress caused by drought or salt injury (Liu et al., 1998; Kasuga et al., 1999). CBF overexpression in transgenic Brassica napus and tobacco also increased their drought and low temperature tolerance (Jaglo et al., 2001; Kasuga et al., 2004). Overexpression of the apple gene MdDREB76 in tobacco conferred salt and drought stress tolerance (Sharma et al., 2019). Overexpressing the Jatropha curcas gene JcCBF2 in Nicotiana benthamiana improved their drought tolerance (Wang et al., 2020).

Overexpression of AmCBF1, cloned from Ammopiptanthus mongolicus, an evergreen, broad-leaved shrub from the northwestern desert of China with great resistance to cold and aridity, significantly enhanced cold tolerance, reduced electrolytic leakage, and increased soluble sugar and proline content in transgenic tobacco compared to the wild type (Gu et al., 2013). However, the application of AmCBF1 in regulating cotton resistance to drought and cold stress has not been extensively studied. In this study, we introduced AmCBF1 into cotton using Agrobacterium tumefaciens to produce independent transgenic lines. Further analysis of two transgenic lines showed that AmCBF1 overexpression increased the seed germination rate, root length, and other physiological indices during drought and cold treatment, and the photosynthetic ability of transgenic plants was significantly higher than in the wild type. Our results indicate that this new transgenic cotton germplasm with enhanced stress resistance and photosynthetic capacity is suitable to be used for cultivation in northwest China.

Materials and Methods

Cotton transformation

The full-length cDNA of the AmCBF1 (GenBank accession no. EU840990.1) gene was amplified and integrated into the pBI121 vector between the Xba I and BamH I restriction sites. The recombinant plasmid, pBI121-AmCBF1, was introduced into the Agrobacterium tumefaciens strain LBA4404 (Gu et al., 2013) to transform upland cotton variety R15 using the previously described protocol (Wang et al., 2017). After germination, the regenerated plants with true leaves and roots were transferred to soil.

Molecular identification

Genomic DNA was extracted from young leaves of transgenic cotton using the DNA secure plant kit (Tiangen, Beijing, China). PCR analysis was performed to detect AmCBF1 in transgenic plants using the specific primer pair AmCBF1-F and AmCBF1-R (Table S1) and the thermal cycling conditions (PCR thermal cycler, Gene Company, Shanghai, China) of 94 °C for 5 min; 35 cycles of 94 °C for 30 s, 58 °C for 30 s, and 72 °C for 60 s; and 72 °C for 10 min. PCR products were separated using 1% (w/v) agarose gel electrophoresis and visualized with an ultraviolet spectrometer. Semi-quantitative reverse transcription PCR (RT-PCR) and quantitative real time PCR (qRT-PCR) were carried out to analyze the relative expression of AmCBF1 in R15 and transgenic cotton. Total RNA was extracted using the Plant RNA Kit (Yuanpinghao, Beijing, China) and cDNA was synthesized using the TransScript One-Step gDNA Removal and cDNA Synthesis SuperMix Kit (TransGen Biotech, Beijing, China) according to the manufacturer’s instructions. The specific primer pair AmCBF1-F and AmCBF1-R was used for RT-PCR. The PCR amplification protocol was as follows: 94 °C for 5 min; 30 cycles of 94 °C for 30 s, 58 °C for 30 s, and 72 °C for 60 s; and 72 °C for 10 min. For qRT-PCR, the primers qA-F and qA-R were used. The amplifications were carried out at 95 °C for 1 min; 40 cycles of 95 °C for 15 s, and 60 °C for 30 s. Gene expression data was evaluated using the 2−ΔΔCt method (Livak & Schmittgen, 2001). Small subunit ribosomal RNA (SSU) (GenBank accession number U42827) was used as the internal reference gene. The primer details are listed in Table S1. The lines showing the presence of transgenes were selected and propagated. T4 generation seeds were used for further study.

To uncover the insertion site, whole-genome resequencing was performed by the LiBaijia company (Beijing, China). The high-quality clean data were mapped to the reference genome of upland cotton (https://cottonfgd.org/) (Zhu et al., 2017) using BWA software based on the resequencing results. To further confirm the exact insertion site of AmCBF1 in upland cotton, primers were designed for the pBI121-AmCBF1 plasmid and the upland cotton genome at both sides of the insertion site according to the results of whole-genome resequencing. The primers are listed in Table S1.

Phenotype analysis of transgenic cotton

To measure plant root length, shoot height, and root and shoot dry mass, whole R15 and transgenic cotton plants grown in the greenhouse were sampled randomly at the two-leaf stage. Averages were calculated for at least six separate individuals per line.

For seed germination analysis, transgenic lines L28, L30, and upland cotton variety R15 seeds were placed on sterile filter paper in a petri dish supplemented with water or mannitol (300 mM). The control and drought stress groups were cultured at 25 °C for 7 days, and the low temperature treatment group was placed in a climate chamber at 15 °C for 7 days. The germination rate was scored, and the radicle length of the seedlings was measured after 7 days. Thirty seeds were selected per line each time. The experiment was repeated four times.

Physiological analysis under drought and cold stress

The T4 generation seeds of the homozygous transgenic cotton (L28 and L30) and R15 lines were sowed in pots containing a 1:1 mixture of peat and vermiculite in the greenhouse with 14 h light at 28 °C/10 h dark at 20 °C. When the two cotyledons of the seedlings fully expanded, the plants were divided into three groups: typical watering group (control), drought treatment group (300 mM mannitol), and cold treatment group (4 °C). The control plants were irrigated with water two times per week. For drought treatment, the plants were irrigated twice weekly with 300 mM mannitol for 30 days. For the cold stress, the plants were treated at 4 °C for 48 h. Following treatment, the physiological traits of the seedlings were measured as previously described: relative water content (RWC) of the leaves (Yue et al., 2012), electrolyte leakage from membranes in leaves (Gu et al., 2013), chlorophyll content (Arnon, 1949), and soluble sugar (Lv et al., 2007). Each cotton line under stress treatment included at least three replicates (pots).

Measurements of lower epidermal cell area, stomatal density, and photosynthesis

We washed the fourth true leaf from the R15 and transgenic cotton lines (L28 and L30) grown in the greenhouse at the five-leaf stage with water and treated them with 2.5% glutaraldehyde for 8 h at 4 °C. The lower epidermal cell area and stomatal density of the samples were observed for each line using a scanning electron microscope. The protocol was used as previously described (Beaulieu et al., 2008). The epidermal cell area was measured using Image J software (National Institutes of Health, Bethesda, MD, USA). Transgenic cotton and R15 lines were planted and divided into three groups as previously described. Each cotton line under each stress treatment consisted of at least three replicates. The photosynthesis of the fourth true expanded leaf in the control, drought, and cold treatment groups (15 °C for 48 h) was assessed using a portable infrared gas analyzer according to the instructions (Li-6400, Li-Cor Inc., Lincoln, NE, USA). Net photosynthesis, stomatal conductance, intercellular CO 2concentration, and transpiration rate were measured on a clear warm day. The saturating photosynthetic photon flux density (PPFD) was maintained at 1,500 µmol m−2 s−1 emitted from a red-blue (665 and 470 nm, respectively) LED light source attached onto the leaf measurement chamber. The CO2 concentration flowing into the leaf chamber was controlled at 400 µmol mol−1. The cover area of the leaves was set as six cm2. Leaf temperature was set at 25 °C, and the humidity was set to 50–70% for all measurements (Tricker et al., 2005).

Statistical analysis

All the assays above were repeated at least three times with three biological replicates. The means of three independent experiments were analyzed for significant differences using Duncan’s multiple range tests and SPSS software (IBM SPSS statistics version 21, IBM, Armonk, NY, USA). Results were considered statistically significant when P < 0.05.

Results

Generation and molecular identification of transgenic cotton overexpressing AmCBF1

The T-DNA structure of the pBI121-AmCBF1 and the location of related primers are shown in Fig. 1A. The representative images during the transformation process of cotton are shown in Fig. 1B. The PCR results indicated that 11 independent transgenic cotton lines were obtained after Agrobacterium-mediated genetic transformation (Fig. 1C). Two dominant homozygous lines, L28 and L30, were selected based on PCR results for further analysis. The RT-PCR results showed that the AmCBF1 gene was expressed in both cotton lines but not in R15. qRT-PCR showed that the expression level of AmCBF1 in L30 was 1.73-times higher than that in L28 (Fig. 1D). Comparing whole-genome resequencing data with the upland cotton reference genome indicated that there were 10 and 22 pairs of matched reads separately on chromosomes A05 and A12, respectively, in L28, and 24 and 11 pairs of matched reads separately on chromosomes A12 and A13, respectively, in L30. The whole-genome resequencing showed that the insertion sites of L28 and L30 were located at 8998563–8998590 on chromosome A05 and 4379624–4379637 in chromosome A13, respectively (Fig. S1). Three pairs of different primers were designed to verify the left and the right borders of the insertion sites on chromosome A05 and A13 separately. The PCR results further confirmed the insertion site that was determined using genome resequencing (Fig. 1E). Intriguingly, the resequencing data indicated that both L28 and L30 contained some matched reads on chromosome A12, and the insertion site was almost always located at 86125018–86125479. Three pairs of different primers were designed to evaluate the insertion site on chromosome A12. The results showed that the specific target fragment amplified in L28 or L30 could also be obtained in R15 (Fig. S2). Further sequencing analysis revealed an endogenous GhDREB (GenBank accession No. XM_041083773.1) sequence existed in this region. The homology between the mRNA sequence of this GhDREB with AmCBF1 was 57.37%, and the core sequence of this GhDREB with AmCBF1 was 77.08% (Fig. S3). The amino acid sequence homology between this GhDREB and AmCBF1 was 58.56% (Fig. S4). These results proved that there was no T-DNA insertion in chromosome A12 of L28 and L30, and they were both single-copy insertion.

Figure 1 Transgenic cotton plant regeneration and molecular identification.

(A) Schematic of pBI121-AmCBF1 gene expression vector. The AmCBF1 gene was controlled by CaMV35S promoter. (B) Steps in Agrobacterium-mediated transformation. (1), induction of resistant calluses; (2), resistant callus induction and proliferation subculture; (3), embryogenic callus induction; (4), plant regeneration; (5), transplant regenerated transgenic cotton. (C) Agarose gel of PCR products using primers specific to the AmCBF1 gene from AmCBF1-transgenic plants. Lane M, DNA maker; R15, wild type; P+: positive control; L1–11, AmCBF1-overexpression lines. (D), Relative expression of AmCBF1 in wild type and transgenic lines as determined by qRT-PCR and RT-PCR. (E), PCR verification of the left and right flanking sequence in transgenic cotton lines (L28 and L30). Lane M, DNA Maker; L, the PCR product of the left flanking sequence; R15, wild type; N, negative control; R, the PCR product of right flanking sequence.

AmCBF1 participation in plant growth regulation

Phenotypes were examined under normal growth conditions when the first two true leaves of cotton had fully expanded (Fig. 2A). The root length of L28 (about 12.63 cm) was significantly longer than that of L30 (10.58 cm). The difference in root length of L28 and L30 was not significant in comparison to R15 (11.92 cm). The average plant height of L28 (13.51 cm) and L30 (12.41 cm) was significantly shorter than that of R15 (17.50 cm) (Fig. 2B1). The root to shoot ratio of the transgenic seedlings (more than 0.10) was significantly higher than that of R15 (0.08) (Fig. 2B2).

Figure 2 Phenotype and stress analysis.

(A) Growth phenotype of R15 and L28, L30 in the greenhouse with 14 h light at 28 °C/10 h dark at 20 °C. (B) Characteristics of phenotype in transgenic plants and R15. (1) Root and shoot length. (2) Root to shoot ratio. (C) The state of seed germination under normal conditions and stress treatments. (D) (1) Seed germination rate, (2) root length of R15 and transgenic lines after 7 days of growth conditions. Values are means ± SE of at least three replications. Different letters above the columns indicate significant differences between R15 and transgenic lines in the control or stress treatments (P < 0.05).

Seed germination and root growth under drought and cold stress

Seed germination under different treatments was observed and recorded. In the control group, there were no differences in the germination rates among the R15 and two transgenic lines 7 d after sowing at 25 °C. Germination was delayed by 1 or 2 days across all seeds under drought or 15 °C conditions. The germination rate of the R15 seeds was 78.33% in drought and 2.50% in 15 °C treatments, which was significantly lower than that of L28 (97.50% in drought, 41.67% in cold) and L30 (100% in drought, 71.67% in cold) (Figs. 2C and 2D1). There was no significant difference in the root length of R15 and transgenic cotton 7 d after sowing at 25 °C. However, after treatment with 300 mM mannitol, the root length of R15 was about 2.08 cm, compared with 2.60−2.68cm in the transgenic lines. Similarly, the root length of R15 was 0.20 cm compared with 0.42 cm of L28 and 0.49 cm of L30 following cold treatment. The root lengths of these two transgenic lines were always significantly longer than R15 under drought or cold stress (Figs. 2C and 2D2). Although root growth is expected to be inhibited at low temperatures, it was less inhibited in the transgenic plants than in R15. These results suggested that overexpression of AmCBF1 gene in cotton can improve drought and cold tolerance during the seed germination stage.

Physiological analysis under drought and cold stress

To determine whether AmCBF1 increased the resistance of transgenic cotton plants to drought or cold stress, multiple physiological indicators were measured before and after drought and low temperature treatments. The RWC in transgenic and R15 plants was more than 80% under normal conditions, and always significantly higher in transgenic lines than in the wild types. Compared with the control group, the RWC of transgenic cotton and R15 decreased under drought or cold stress. The RWC in L28 (about 80.23%) and L30 (about 84.08%) were significantly higher than in R15 (72.55%) after drought treatment. Under cold treatment, the RWC in the two transgenic lines (over 70%) were significantly higher than in R15 (65.52%) (Fig. 3A). Electrolytic leakage in the leaves of transgenic cotton and R15 were around 25% under normal conditions, but were up to 39.48% in R15, 29.71% in L28, and 28.72% in L30 under drought stress. Under cold treatment, the electrolyte leakage reached 56.15% in R15 and less than 30% in the two transgenic lines. These two transgenic cotton lines showed notably less electrolyte leakage compared to R15 under drought or cold stress (Fig. 3B). AmCBF1 appeared to confer protection against membrane leakage in the transgenic lines. The chlorophyll content in transgenic cotton was always significantly higher than in R15. The average chlorophyll content decreased after 300 mM mannitol or low temperature treatment. It was reduced by 22.25% (R15), 21.86% (L28), and 18.70% (L30) after drought treatment, and 31.21% (R15), 27.28% (L28), and 20.18% (L30) after cold treatment. The decreasing trend of chlorophyll content was more moderate in transgenic cotton than in the control group (Fig. 3C). The content of soluble sugars in the two transgenic lines was significantly higher than in the wild type under normal conditions (1.94 times higher in L28 and 2.22 times higher in L30). L28 was 1.60 times higher and L30 was 2.37 times higher than R15 under drought stress. L28 was 2.10 times higher and L30 was 1.73 higher than R15 under low temperature stress. Drought and low temperature stress significantly increased the soluble sugar content in both R15 and the transgenic lines compared with the controls (Fig. 3D). These results further suggested that AmCBF1 overexpression conferred drought and cold tolerance to cotton.

Figure 3 Analysis of physiological indicators in R15 and transgenic plants.

(A) Relative water content. (B) Electrolyte leakage. (C) Chlorophyll content. (D) Soluble sugar content. Values are means ± SE of at least three replications. Different letters above the columns indicate significant differences between R15 and transgenic lines in the control or stress treatments (P < 0.05).

Measuring low epidermal cell area, stomatal density, and photosynthesis in cotton seedlings

The epidermal cells were usually irregular or polygonal in R15, but they were mostly rectangular and arranged regularly and tightly in the transgenic lines (Fig. 4A). The transgenic lines also had a markedly smaller area of epidermal cells and higher stomatal density on leaves than R15 (Fig. 4B). Net photosynthesis, stomatal conductance, intercellular CO2 concentration, and transpiration rate in the transgenic cotton lines were all significantly higher than R15 under normal conditions (Fig. 4C). When compared with those under normal conditions, net photosynthesis, stomatal conductance, and transpiration rate were dramatically inhibited by drought and cold treatments, but were all significantly higher in the transgenic lines than in R15. Intercellular CO2 concentration was higher in transgenic cotton than in R15 before and after drought treatment, but there was no difference under cold stress (Fig. 4C). AmCBF1 overexpression improved leave photosynthesis of cotton seedlings, which may be due to AmCBF1 regulating epidermal cell differentiation and formation and improving photosynthesis-related parameters, such as stomatal density.

Figure 4 Microstructure analysis of epidermal cells and photosynthetic performance of R15 and transgenic cotton lines.

(A) Scanning electron micrograph of epidermal cells of cotyledons from R15 and transgenic L28 and L30. (B) Area of epidermal cells and stomatal density of R15 and transgenic lines L28 and L30. (C) Net photosynthesis, stomatal conductance, intercellular CO2 concentration, and transpiration rate. Plants were treated with water (control), mannitol, or 15° C. The values are means ± SE of at least three replications. Different letters above the columns indicate significant differences between R15 and transgenic lines in the control or stress treatments (P < 0.05).

Discussion

In this study, the AmCBF1 gene was successfully introduced into R15 via Agrobacterium mediation. In our functional analysis of two homozygous transgenic cotton lines, the overexpression of AmCBF1 enhanced the resistance to drought and cold stresses compared to R15 plants.

A plant’s root system is an important functional organ. It directly interacts with soil in capturing soil resources, but is severely constrained by environmental stresses, both biotic and abiotic (Yildirim et al., 2018; Zobel, 1995). To adapt and respond to diverse environments, plant roots have developed unique capabilities to sense and respond to nutrients in soil (Kazan, 2013). In general, the distribution, production, and accumulation of plant roots are related to water availability in the soil. Well-developed and robust root systems are positively correlated with drought resistance. Root length, root dry mass, and root to shoot ratio are important indicators of drought resistance. Root to shoot ratio is a sensitive indicator and important index used to evaluate a plant’s health and capacity to take up light, water, and nutrients (Agathokleous et al., 2018; Luo et al., 2013). Increasing the root to shoot ratio for length and mass could improve the plant’s ability to take up water, maintain a high-water status, and be protected from drought stress (Zhang et al., 2018). Changes in root to shoot ratio may be related to the distribution of carbohydrates between shoots and roots (Xu et al., 2015). When determining morphological traits in this study, the higher root to shoot ratio in transgenic cotton lines compared to R15 under normal control conditions implied that AmCBF1 significantly reduced growth and may participate in energy metabolism pathways. Previous research has shown that root to shoot ratio is negatively correlated with plant stature (Li, Luo & Lu, 2008). Increasing root to shoot ratio and decreasing plant stature are the main long-term plant adaptations to cold climates (Körner, 2003). Additionally, electrolyte leakage, malondialdehyde, free proline, and soluble sugar are all important indicators of protection against drought or cold stress and have been widely substantiated (Yue et al., 2012; Gu et al., 2013; Zhang et al., 2014). Drought and cold stress can damage the integrity of cellular membranes, causing electrolyte leakage and accumulation of macromolecules. In our study, transgenic plants had less membrane damage and higher soluble sugar under drought or cold stress. Therefore, various results indicate that AmCBF1 overexpression in cotton improves resistance to drought or cold stress and may alter the partitioning of assimilates between above- and below-ground organs.

The stomatal density of leaves is an important physiological variable for photosynthesis and transpiration because the stomata regulate the exchange of CO2, water vapor, and other constituents between plants and the external atmosphere (Royer, 2001). It has a positive relationship with stomatal conductance and can be altered with the initiation of stomata or various environmental factors (such as CO2 concentration, water stress, and temperature) (Xu & Zhou, 2008; Royer, 2001; Woodward & Kelly, 1995; Beerling & Woodward, 1996). In turn, it can strongly influence water-use efficiency in plants (Woodward & Kelly, 1995). Increased stomatal density enhanced leaf photosynthetic capacity by modulating gas diffusion in Arabidopsis (Tanaka et al., 2013). Smaller stomata are associated with high stomatal densities, which rapidly increase leaf stomatal conductance and maximize CO2 diffusion into the leaf (Zhang et al., 2018). In this study, we noticed a significant difference in the shape and arrangement of leaf epidermal cells between transgenic cotton and wild type, as well as the area of epidermal cells and the stomata density (Figs. 4A and 4B). The results revealed that the AmCBF1 gene might directly or indirectly participate in epidermal cell and stomatal differentiation and formation, and change the proportion of epidermal cells that differentiate into guard cells and stomata. These microscopic changes in cells may be the key factor in improving plant stress resistance. Greater stomatal density, as seen in the transgenic plants compared with the wild type, coupled with the greater capability to regulate transpiration rate, characterize a drought-resistant cultivar (Ferreyra et al., 2000).

Assessing variability in net photosynthesis and stomatal conductance has been suggested for selecting genotypes with higher drought tolerance (Lv et al., 2007). The decrease in net photosynthesis caused by drought stress is due to the closure of stomata, which restricts the diffusion of CO2 into the leaf, or a decrease in the photosynthetic capacity of mesophyll cells (Lv et al., 2007; Farquhar & Sharkey, 1982). In this study, the net photosynthesis in transgenic cotton seedlings was much less inhibited by drought and cold stress than in R15 (Fig. 4C). This indicated that the overexpression of AmCBF1 in cotton improved gas exchange, and may be due to the higher stomatal density creating higher stomatal conductance of the transgenic cotton leaves. After drought treatment, the stomatal conductance of the transgenic cotton plants decreased, while the intercellular CO2 level increased in leaves. This result indicated that the decrease in the net photosynthetic rate of the plant after drought stress mainly depended on the photosynthetic capacity of the mesophyll cells, and is further evidence that stomatal conductance is the main factor limiting photosynthetic rate.

In summary, inserting AmCBF1 markedly improved transgenic cotton’s tolerance to drought and cold stress, and enhanced the leaf photosynthetic capacity. The transgenic line holds promise as a germplasm resource used to enhance multi-stress tolerance in cotton breeding. Further studies are required to determine the field performance of transgenic cotton, especially in the Xinjiang area.

Supplemental Information

Supplemental Information 1 The insertion sites of AmCBF1 gene were localized on chromosomes A05 in L28 and A13 in L30 separately

The whole-genome resequencing showed that insertion sites of L28 and L30 were located at 8998563–8998590 on chromosome A05 and 4379624–4379637 in A13 chromosome, respectively.

Click here for additional data file.

Supplemental Information 2 PCR analysis of A12 flanking sequence in AmCBF1-transgenic cotton plants

Three pairs of primers amplify the same size products in R15 and transgenic plants with the same sequence. A, Agarose gel of PCR products using primers A12-1F and A12-1R. B, Agarose gel of PCR products using primers A12-2F and A12-2R. C, Agarose gel of PCR products using primers A12-3F and A12-3R. Lane M, DNA marker; L28, transgenic cotton L28; L30, transgenic cotton L30; R15, wild type. N, negative control.

Click here for additional data file.

Supplemental Information 3 Comparison of identity between AmCBF1 and the endogenous Gh DREB sequence at the insertion site of A12 chromosome

Sequencing analysis of the endogenous Gh DREB and AmCBF1 gene. The homology between the mRNA sequence of this GhDREB with AmCBF1 was 57.37%, The homology of the partial zone of this GhDREB with AmCBF1 (in the red box) reached 77.08%.

Click here for additional data file.

Supplemental Information 4 Analysis of the amino acid sequence of the of GhDREB on A12 chromosome with AmCBF1.

The amino acid sequence homology of the of GhDREB on A12 chromosome with AmCBF1 was 58.56%.

Click here for additional data file.

Supplemental Information 5 Primers used in this study

Click here for additional data file.

Supplemental Information 6 Rata data

Click here for additional data file.

Additional Information and Declarations

Competing Interests

Author Contributions

Data Availability

The authors declare there are no competing interests.

Guoqing Lu performed the experiments, analyzed the data, prepared figures and/or tables, authored or reviewed drafts of the paper, and approved the final draft.

Lihua Wang performed the experiments, prepared figures and/or tables, and approved the final draft.

Lili Zhou performed the experiments, prepared figures and/or tables, and approved the final draft.

Xiaofeng Su analyzed the data, authored or reviewed drafts of the paper, and approved the final draft.

Huiming Guo conceived and designed the experiments, performed the experiments, analyzed the data, authored or reviewed drafts of the paper, and approved the final draft.

Hongmei Cheng conceived and designed the experiments, authored or reviewed drafts of the paper, and approved the final draft.

The following information was supplied regarding data availability:

The raw data of measurements is available in the Supplementary Files.

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
