# Peer review of "Overexpression of AmCBF1 enhances drought and cold stress tolerance, and improves photosynthesis in transgenic cotton"

_PeerJ, doi:10.7717/peerj.13422_

## Round 0.1 · original submission · Major Revisions

Please address all the comments raised by the Reviewers.

Reviewer 1 ·

Basic reporting

This work needs more experimental work to validate the transgenic materials generated, before doing the phenotyping analyses. Therefore, I have to reject it in its current form.

Experimental design

This work needs more experimental work to validate the transgenic materials generated, before doing the phenotyping analyses. Therefore, I have to reject it in its current form.

Validity of the findings

This work needs more experimental work to validate the transgenic materials generated, before doing the phenotyping analyses. Therefore, I have to reject it in its current form.

Reviewer 2 ·

Basic reporting

In this research AmCBF1 was isolated and transformed into upland cotton cultivar R15 to evaluate potential benefits. The results showed that transgenic plants had distinctly higher relative water content, chlorophyll content, soluble sugar content and lower ion leakage after drought and cold stress. The area of lower epidermal cells, stomatal density and root to shoot ratio varied significantly between transgenic cotton lines and R15. Photosynthetic ability in transgenic plants was inhibited after stress, net photosynthetic rate, stomatal conductance and transpiration rate in transgenic plants were significantly higher than in R15. The results showed that the enhanced stress tolerance of transgenic cotton achieved by improving photosynthesis. The new transgenic cotton germplasm has the great application value to deal with the abiotic stress.

Experimental design

1. Phenotype analysis of transgenic cotton was measured at the two-leaf stage, while lower epidermal cellarea, stomatal density, and photosynthesis were measured at the five-leaf stage of the fourth true leaf. Please clarify the reason for these two different stage and compare the effects on the results.
2. Regarding the plants cold treatment, I noticed that sometimes 4°C cold stress is used, and sometimes 15°C cold stress is used. Please discuss the difference between the two different temperature treatments.
3. If possible, transgenic cotton with 3 lines will much better than 2 lines of 28 and 30.

Validity of the findings

The manuscript is well written and the results, discussion and conclusion are well supported by the sufficient experimental evidences. The study is interesting and has informative data on cotton, an important commercial crop.

Additional comments

Please briefly introduce the CBF1 homologous genes in cotton endogenous.

Reviewer 3 ·

Basic reporting

In the present study, authors have transformed AmCBF1 gene from the shrub Ammopiptanthus mongolicus into upland cotton cultivar R15 and characterized the role of this gene in response to drought and cold stress. They genotyped the transgenic lines and verified the copy number of the transgene by sequencing the genome of two selected transgenic lines. They studied different physiological parameters associated with photosynthesis during the control and stress conditions in R15 and the two transgenic lines. They observed a higher rate of photosynthesis in overexpression lines of cotton in comparison to R15.

Authors have provided a good literature background and included all the raw data of their experiments. They performed all the necessary experiments to support their hypothesis.

The authors have made a good attempt to describe their experimental results but I would highly recommend them to describe the experimental methods briefly and have a thorough grammatical and spell-check for the entire manuscript. The figure legends also need to include the abbreviations.
Details are provided in the comment section.

In my opinion, the major concern of this study is that they have identified a copy of DREB in the selected transgenic cotton lines which had 77.08% similarity with the AmCBF1 gene. But it is not clear whether they checked the sequence at the mRNA and protein level. Hence it would be good to include the transgenic lines expressing the empty vector, as a control in their study. Using this will help rule out the possibility of any changes associated with the transformation process.

Comments and suggestions:

Title and Introduction:

1. Replace “enhance” with “enhances”

2. Rephrase the sentence from line 11 to 16 in the Abstract section as: “AmCBF1 (Ammopiptanthus mongolicus C-repeat-binding factor) was isolated from the shrub Ammopiptanthus mongolicus, and inserted into upland cotton (Gossypium hirsutum L.) cultivar R15 to evaluate the potential benefits of this gene. Two transgenic lines were selected and the transgene insertion site was identified by whole-genome sequencing. The results showed that AmCBF1 was incorporated into the cotton genome as a single copy”.

3. Rephrase the sentence from line 33-38. It has been mentioned that northwest inland area is one of the largest producers of cotton and at the same time it is also mentioned that it has recently become the largest cotton-growing and producing area. So rephrase the sentence.

4. In line 67 and 68 mention the source of the gene from which the gene was isolated. For example, “the apple gene MdDREB76”

Methods:

1. After germination of the transgenic lines, why were they grafted to a different cotton variety? Is there a significance of this process? Fig 1B. showing the direct transplantation of the transgenic lines onto the soil. Please check and include the necessary corrections.

2. Line 102: Provide the link for download of genome sequence of upland cotton variety and also include the reference

3. It would be good to include a brief schematic of genotyping of transgenic cotton lines, describing the location of the different primers used in the study.

4. Write as “supplementary table or figure” wherever applicable instead of “Supplement “

5. Quantitative real-time PCR was done using the cDNA of the transgenic lines so please replace the term “qPCR” with qRT-PCR wherever applicable

6. Please mention the generation of transgenic lines used for the phenotypic and qPCR analysis in material and methods section

7. Rewrite line 117 as “For seed germination analysis, transgenic lines L28, L30 and upland cotton variety R15”

8. For the physiological analysis of cold stress in transgenic cotton lines, the temperature used for the study was mentioned differently as 4°C but for other studies a temperature of 15°C was used. I’m curious why different temperatures were used?

9. Provide proper description of the methods used to measure photosynthesis rate, stomatal conductance etc. and also include the appropriate references in the text. For example, in line 144-145, reference is not included for measuring net photosynthesis and other parameters. Also, briefly describe the SEM method too.


Results: Results need to be described with little more details.

1. For example, the homology of DREB with AmCBF1 was compared but nowhere it was mentioned as whether it was the gene or mRNA sequence.
2. In the section from line 177, describe the control results as well. The difference in root length was not significant in comparison to R15
3. In the Fig 1D, mention as relative expression of AmCBF1
4. For all the figure legends, please mention any abbreviations used. For example, in Fig 1, please describe P+, L, N, R
5. In Fig 1D, what is SSU?
6. For Fig 1E, mention the primer sets used for genotyping. It can be included either in the methods section or in the figure legends
7. In discussion section, the sentence in line 260, “the relatively shorter shoot height and ……than that of R15” is not needed.
8. Spell check line 299
9. Fig S3, A12 insertion sequence was shown but the transgenic line was not mentioned, whether L28 or L30? It is good to include some details in the figure legends or in the manuscript
10. Crosscheck the references and do a thorough spell-check

Experimental design

The experimental design and the number of replicates is appropriate. However authors have to describe the methods in little detail and provide appropriate references wherever applicable. Details included in the comments section.

Validity of the findings

Authors have conducted all the necessary experiments to support their hypothesis. They have performed the statistical analysis to show the significant differences between the wild type and transgenic cotton lines under control and stress conditions.

---

## Round 0.2 · Minor Revisions

Please download the annotated PDF file and address the issues raised by the Reviewer No. 3. Looking forward to receiving your revised manuscript soon.

Reviewer 2 ·

Basic reporting

no comment

Experimental design

no comment

Validity of the findings

no comment

Additional comments

no comment

Reviewer 3 ·

Basic reporting

Authors have improved the manuscript by incorporating all the suggestions and comments. However, there are minor revisions needed and the same are included in the edited version uploading here.

Experimental design

As per the suggestions, the authors have described the experimental methods in detail. However, the figure legends should be included either in the figures itself or as a separate file with all the relevant details and abbreviations.

Validity of the findings

All are results are well described.

Annotated reviews are not available for download in order to protect the identity of reviewers who chose to remain anonymous.

---

## Round 0.3 · Minor Revisions

The Section Editor identified some issues that need to be addressed before the manuscript can be accepted:

> The work is good but I disagree with the conclusion that the enhanced stress tolerance is _caused_ by improved photosynthesis (title and abstract).
>
> Causality has not been established and it is just as likely (or even more so) that improved photosynthesis is a consequence of improved stress tolerance. For example, the higher root to shoot ratio in the transgenic lines might lead to better water acquisition, allowing improved photosynthesis.
>
> I am not saying that this is the cause either, just pointing out that there other possibilities.
>
> So, the authors should rewrite the title and abstract and any other parts of the paper to remove the idea that the tolerance is caused by increased photosynthesis.
>
> Also there is a typo in the x axis legend of figure 3B "Ccontrol"

In addition, please specify the type of stresses in the Title. Also, a thorough check of English and a subsequent improvement in English quality is suggested.

---

## Round 0.4 · accepted · Accept

Congratulations! Your manuscript has been accepted for publication in PeerJ.

Reviewer 3 ·

Basic reporting

The authors have addressed all the minor revisions suggested and the manuscript can be accepted for publication.

Experimental design

No comments

Validity of the findings

No comments